# Ultrasonic Extraction of Bioactive Compounds from Green Soybean Pods and Application in Green Soybean Milk Antioxidants Fortification

**DOI:** 10.3390/foods11040588

**Published:** 2022-02-18

**Authors:** Noppol Leksawasdi, Siraphat Taesuwan, Trakul Prommajak, Charin Techapun, Rattanaporn Khonchaisri, Nattha Sittilop, Anek Halee, Kittisak Jantanasakulwong, Suphat Phongthai, Rojarej Nunta, Maneerat Kiadtiyot, Arisa Saefung, Julaluk Khemacheewakul

**Affiliations:** 1Agro-Bio-Circular-Green and Bioprocess Research Cluster, Faculty of Agro-Industry, School of Agro-Industry, Chiang Mai University, Chiang Mai 50100, Thailand; noppol.l@cmu.ac.th (N.L.); siraphat.t@cmu.ac.th (S.T.); charin.t@cmu.ac.th (C.T.); jantanasakulwong.k@gmail.com (K.J.); su.phongthai@gmail.com (S.P.); 2Division of Food Science and Technology, Faculty of Agro-Industry, School of Agro-Industry, Chiang Mai University, Chiang Mai 50100, Thailand; rattanaporn.khon@gmail.com (R.K.); nattha.sittilop@gmail.com (N.S.); jennymaneerat11@gmail.com (M.K.); pedfung48@gmail.com (A.S.); 3Cluster of Innovative Food and Agro-Industry, Chiang Mai University, Chiang Mai 50100, Thailand; 4Division of Food Safety, School of Agriculture and Natural Resources, University of Phayao, Phayao 56000, Thailand; tpromjak@gmail.com; 5Division of Food Science and Technology, Faculty of Science and Technology, Kamphaeng Phet Rajabhat University, Kamphaeng Phet 62000, Thailand; nek_ha@hotmail.co.th; 6Division of Packaging Technology, Faculty of Agro-Industry, Chiang Mai University, Chiang Mai 50100, Thailand; 7Division of Food Science and Technology, Faculty of Science and Technology, Lampang Rajabhat University, Lampang 52100, Thailand; quan_rn@hotmail.com

**Keywords:** green soybean, ultrasonic extraction, bioactive compounds, antioxidant, dairy product, fortification, pod, *Glycine max* L.

## Abstract

Green soybean (*Glycine max* L.) pods (GSP) are agro-industrial waste from the production of frozen green soybean and milk. These pods contain natural antioxidants and various bioactive compounds that are still underutilized. Polyphenols and flavonoids in GSP were extracted by ultrasound technique and used in the antioxidant fortification of green soybean milk. The ultrasound extraction that yielded the highest total polyphenol content and antioxidant activities was 50% amplitude for 10 min. Response surface methodology was applied to analyze an optimum ultrasonic-assisted extraction (UAE) condition of these variables. The highest desirability was found to be 50% amplitude with an extraction time of 10.5 min. Under these conditions, the experimental total phenolic content, total flavonoid content, and antioxidant activity were well matched with the predicted values (R^2^ > 0.70). Fortification of the GSP extracts (1–3% *v*/*v*) in green soybean milk resulted in higher levels of bioactive compounds and antioxidant activity in a dose-dependent manner. Procyanidins were found to be the main polyphenols in dried GSP crude extracts, which were present at a concentration of 0.72 ± 0.01 mg/100 g. The addition of GSP extracts obtained by using an ultrasound technique to green soybean milk increased its bioactive compound content, especially procyanidins, as well as its antioxidant activity.

## 1. Introduction

Free radicals shorten the shelf-life of food products as well as increase oxidative stress within cells, a recognized pathologic pathway of several chronic diseases [1]. During the past two decades, the utilization of either bioactive compounds or natural antioxidants in food products or by-products through both non-biological [2] or biological means [3,4,5,6] has gained considerable interests due to the participating roles of these compounds in absorption and neutralization of free radicals, thereby slowing down the autoxidation process. The benefits of natural antioxidants were due not only to their biological values, but also to their economic impact, as most of them can be extracted from food by-products and under-exploited plant species [7].

Legumes are an excellent source of bioactive compounds with antioxidant capacity such as flavonoids, anthocyanin, and other phenolic compounds. Recent studies showed that a high intake of legumes protected the human body against oxidative damage and reduced the risk of type 2 diabetes mellitus [8]. Green soybean (*Glycine max* L.) is a legume that has been shown to exhibit strong antioxidant activity. Peiretti et al. [9] stated that black and green soybean exhibited comparatively higher FRAP values than yellow soybean. Hence, many researchers try to use green soybean as an alternative supplement source of bioactive compounds in food products such as butter cake [10] and cookies [11]. Green soybean pods (GSP), a by-product of green soybean processing, potentially contain natural antioxidants, but research on the antioxidant capacity of these GSP is still limited.

Ultrasonic assisted extraction (UAE) has been widely used to extract antioxidants from plants. High-frequency ultrasonic waves induce contraction and expansion cycles that subsequently cause cavitation, breakage of plant cell walls, and infiltration of solvents into cells. The extraction rate and yield of UAE are influenced by several factors, including ultrasonication time and amplitude [12]. Viell et al. [13] compared homogenizer-assisted extraction with UAE for flavonoid content from teff grains (*Eragrostis tef* (Zucc.) Trotter). Under optimal conditions, the total flavonoid content and the antioxidant capacity were significantly higher when UAE was used. Crupi et al. [14] stated that UAE offers efficiency and reproducibility advantages compared to conventional techniques due to their time-saving, ease of procedure, and environment-friendly properties, as well as yielding a cost-effective output of high-quality phenolic extracts. Therefore, UAE could be an alternative green technology over conventional methods (e.g., distillation, maceration, and Soxhlet), which use high temperatures and concentrations of organic solvents [15]

The objectives of the present work were (1) to optimize the UAE process with water as eco-friendly solvent to obtain GSP extracts enriched in antioxidants in the classes of total content of phenolic and flavonoid; (2) to investigate the antioxidant capacity of green soybean milk supplemented with GSP extracts; and (3) to identify some specific antioxidants in the GSP extracts. The novelties of this study were the use of a green technology in the extraction of natural antioxidants from GSP by-products and the demonstration of real-world application in green soybean milk fortification, which will encourage technology adoption by the industrial sector.

## 2. Materials and Methods

### 2.1. Materials

Whole green soybeans (*Glycine max* L.) were obtained from Lanna Agro Industry Co., Ltd. (LACO, Chiang Mai, Thailand). GSPs were separated from whole beans and washed using tap water. The pods were further cut into small pieces of approximately 1 cm squares using a stainless-steel knife and oven-dried at 60 °C for 24 h in a hot-air oven (Memmert UF 110, Schwabach, Germany) until moisture content decreased below 10% [16]. Dried GSPs were ground to fine powder using an electric chopper (Model DPA130, Tefal, France), and the powder was sieved through a 20-mesh sieve. Finally, the dried powder samples were packed in vacuum polyethylene bags and stored at 4 °C before use. All chemicals used were analytical grade.

### 2.2. Ultrasonic-Assisted Extraction

Ultrasonic-Assisted Extraction (UAE) was carried out according to our previous study, Zhou et al. [17], and Sharayei et al. [18], with some modifications. Specifically, 5 g of dried GSP powder was placed in a 250 mL beaker and extracted with 100 mL distilled water using an ultrasonic probe (VX500, Hartford, CT, USA) with a maximum power of 500 W at 20 kHz frequency. The ultrasonication extraction process was carried out in an ice-water bath to prevent heating of samples for three different durations (10, 15, and 20 min) and three different amplitudes (30, 40, and 50%) [19]. The mixtures were centrifuged at 3000× *g* for 15 min at 4 °C (Nüve NF400R, Ankara, Turkey), and the supernatant was filtered through filter paper (Whatman No. 1, Wallingford, UK). The filtered extracts were collected in a centrifuge tube and kept at −18 °C until further analysis.

### 2.3. Preparation of Green Soybean Milk

Fresh green soybean seeds (500 g) were soaked in 2000 mL of tap water for 1 min. The beans were then ground and blended in 1000 mL of drinking water using a blending machine (HR2602, Philips, Ningbo, China) at medium speed until homogeneous (approximately 10 min). The mixture was filtered using a muslin cloth to obtain green soybean milk, to which the GSP extracts were added at the concentration levels of 0 (control), 1, 2, and 3% (*v*/*v*). The green soybean milk was then boiled at 95 °C for 20 min before being immediately placed in an ice bath. The cooled samples were then stored at −18 °C until further analysis.

### 2.4. Determination of Total Phenolic Compounds

Total phenolic compounds were analyzed using the Folin–Ciocalteu method, with some modifications [17]. A 500 μL properly diluted sample or standard solutions of varying concentrations were mixed with 2.5 mL of 1:10 Folin–Ciocalteu:water solution and then thoroughly mixed. After incubation for 8 min at room temperature, 2 mL of 7.5% (*w*/*v*) Na_2_CO_3_ solution was added, and the mixture was immediately mixed and incubated for 2 h. Absorbance was read at 765 nm on a spectrophotometer (G10S UV-Vis, Thermo Fisher Scientific, Waltham, MA, USA). Measurements were conducted in quadruplicate. One mg/mL gallic acid was used as the standard, and the total phenolic compounds of the samples were expressed in mg gallic acid equivalent (GAE) per g sample (mg GAE/g).

### 2.5. Determination of Total Flavonoid Content

Samples (0.25 mL) were mixed with 1.25 mL of distilled water and 75 µL of 5% NaNO_2_ solution and incubated for 6 min at room temperature. The mixture was then combined with 150 μL of 10% AlCl_3_ and 500 μL of 1 M NaOH and brought to 275 mL with distilled water. The solution was mixed thoroughly and left for 5 min at room temperature. Its absorbance was recorded using a spectrophotometer at 510 nm. Catechin equivalents (CAE) per g of sample (mg CAE/g) were used to express total flavonoid contents [20].

### 2.6. Determination of Antioxidant Activity

#### 2.6.1. DPPH Free Radical Scavenging Activity

DPPH (2,2-diphenyl-1-picryl-hydrazyl radical) solution was used to estimate antioxidant activity according to the method described by Lu et al. [21], with minor modifications. Briefly, 150 µL of samples were mixed with 3 mL of 0.6 mM DPPH. The mixture was incubated for 30 min in the dark to allow for complete reaction. Absorbance values of each sample and control (distilled water) were read using a spectrophotometer at 517 nm. The antioxidant activity of each sample was expressed as μmol of Trolox equivalent per g of sample (µmol Trolox/g). 

#### 2.6.2. FRAP Free Radical Scavenging Activity

The Fe^2+^ chelating activity of the samples was measured by the method of Sharma et al. [22], with minor modifications. The ferric reducing antioxidant power (FRAP) reagent was prepared by mixing 25 mL of 0.3 M acetate buffer (pH 3.6/22.8 mM sodium acetate trihydrate) with 2.5 mL of 0.01 M 2,4,6-Tripyridyl-s-Triazine solution and 2.5 mL of 0.02 M FeCl_3_.6H_2_O. An amount of 150 μL of the sample or the control (methanol) was then added to 2850 μL of the FRAP reagent. The reaction mixture was incubated at room temperature in the dark for 30 min, and absorbance was assessed at 593 nm. The antioxidant activity of each sample was expressed as µmol Trolox/g.

### 2.7. Sensory Evaluation of Green Soybean Milk

One hundred untrained panelists (20–40 years of age) were recruited from the Division of Food Science and Technology, Faculty of Agro-industry, Chiang Mai University, Thailand. The samples were coded with a three-digit random number and presented to the panelists. Water was provided for rinsing the mouth between samples. The panelists were asked to provide acceptance scores for color, texture, aroma, sweet taste, salt taste, and overall acceptability based on a standard nine-point hedonic scale (9 = like extremely, 8 = like very much, 7 = like moderately, 6 = like slightly, 5 = neither like nor dislike, 4 = dislike slightly, 3 = dislike moderately, 2 = dislike very much and 1 = dislike extremely) [23].

### 2.8. Quantification of Phenolic Compounds by HPLC

The phenolic compounds (procyanidins, quercetin, glycitin, daidzein, genistin, and linalool) in GSP extracts and green soybean milk were analyzed by high-performance liquid chromatography (HPLC), as previously reported [17,24], with some modifications. Briefly, an Agilent HPLC system (Agilent Technologies, Santa Clara, CA, USA), consisting of a binary pump and a photodiode-array detector equipped with an Agilent Zorbax C18 (4.6 × 250 mm, 3.5 μm) column was employed. The mobile phase consisted of solution A (0.1% *v/v* trifluoroacetic acid) and solution B (pure methanol), which were used to create gradients according to the following program: 0 min, 15% B; 5 min, 25% B; 9 min, 55% B; 12 min, 75% B; 15 min, 75% B; 18 min, 15% B. The flow rate was 0.8 mL/min, and the injection volume was 5.0 µL for procyanidins, quercetin, glycitin, daidzein, and genistin. The program for linalool was 0–20 min, 85% B. The detection wavelength was set at 260 nm for procyanidins, quercetin, glycitin, daidzein, and genistin, and at 210 for linalool. The result was expressed as mg of procyanidins, quercetin, glycitin, daidzein, genistin, and linalool equivalent per 100 g of sample.

### 2.9. Statistical Analysis

Amplitude and exposure time variables were analyzed using two-way analysis of variance for each of the four measures of ultrasonic performance. Comparison among different proportions of GSP supplements in green soybean milk on the antioxidant and sensory characteristic were analyzed using one-way analysis of variance. Significant difference (*p* < 0.05) among samples were followed by Duncan’s new multiple range post-hoc analysis. All of the above analyses were conducted using SPSS for Window version 16. Data were reported as mean values ± standard deviation. Response surface methodology was applied to analyze an optimum UAE condition using Design Expert version 6.0.11 (Stat-Ease, Minneapolis, MN, USA).

## 3. Results

### 3.1. Bioactive Components

Total phenolic contents varied from 85.9 to 107 mg GAE/g across different ultrasonic processing conditions (Table 1). A significantly (*p* < 0.05) higher total phenolic content of 107 ± 0.5 mg GAE/g was obtained using the highest amplitude (50%) for 10 min. Amplitude had a significant effect (*p* < 0.05) on total phenolic content at 10 min extraction time. A higher amplitude creates higher thermal energy to break the plant cellular structure. Increased permeability of cell walls and membranes and the breakdown of secondary metabolites from matrix interactions (polyphenols with lipoproteins) caused enhancement of polyphenols solubility and mass transfer. Thus, a higher ultrasonic amplitude increased extraction efficiency and yielded greater amounts of bioactive compounds [24,25]. Our previous studies also found the same extraction efficiency of phenolics and flavonoid content from green soybean pods, which was achieved using either water or ethanol solution as the extracting solvent (*p* > 0.05). Water was thus indicated to be an efficient solvent in the ultrasound-assisted extraction of green soybean pods. This might be due to the most abundant group of phenolic compounds in the pods being water-soluble. It is worth pointing out that phenolic compounds can be extracted from green soybean pods from a cheap and broadly accessible solvent, which is likewise safe to humans and to the environment. The duration of extraction also influenced polyphenol yields. Increased extraction time from 10 to 20 min at 50% amplitude resulted in significant (*p* < 0.05) reductions in total phenolic content. A longer exposure time could increase solvent temperature beyond optimal levels, resulting in the degradation of thermo-sensitive compounds presented in the GSP samples. Evidently, the extraction condition of 10 min at 50% amplitude was deemed optimal for phenolic content.

Flavonoids have been shown to improve blood lipid profiles; enhance immunity; and have antioxidant, antibacterial, and antitumor properties [26]. Table 1 shows that amplitude and time affected flavonoid content. The highest amount of total flavonoid extracted (8.94 ± 0.1 mg CAE/g) under 50% amplitude was significantly (*p* < 0.05) higher than 30 and 40% within the first 10 min of extraction. In addition, the use of 30–50% amplitude for 15 min extraction time did not show significant difference (*p* > 0.05) in total flavonoid content (8.50–8.75 mg CAE/g) when compared to the maximum concentration of total flavonoid content. However, the extraction efficiency showed a decreasing trend when the extraction time was enhanced from 15 to 20 min (5.75–7.44 mg CAE/g).

When considering the effect of amplitude, a higher amplitude resulted in a higher flavonoid content in the 10–20 min extraction groups. Loss of flavonoids at longer extraction times was due to overheating by the ultrasound treatment, which especially affected the heat-sensitive flavonoids [27]. Flavonoids (e.g., rutin) were more sensitive to thermal degradation than phenolic acids. The concentration of rutin from olive leaves using ultrasound-assisted extraction was 2.11 ± 0.1 mg/g during a longer extraction time of 21 min, which was lower than the extraction time of 7 min (2.22 ± 0.1 mg/g) [28]. According to Bi et al. [29], the gradual increase in the bioactivity of the extract with time may be attributed to the fact that polyphenols, and other bioactive compounds, were still bound within the cell matrices during the early stage of extraction. A sufficient time was thus required to allow for their release. The subsequent decrease in bioactivity might be due to the longer time of exposure to ultrasonic conditions, inducing the degradation or oxidation of these bioactive compounds. Based on these findings, sonication for more than 15 min was found to be unsuitable, as there was not a great amount of total phenolic and flavonoid content extracted by increasing the time interval. It was also clear from the results that the extraction condition of 10 min at 50% amplitude was deemed optimal for both total phenolic and flavonoid content.

### 3.2. Antioxidant Activity

DPPH assay has been used widely and is a popular technique to assess the free radical scavenging activity of different plant extracts. DPPH free radical reduction was determined by the decrease in its absorption at 517 nm when the color of the DPPH assay solution changed from purple to light yellow. The scavenging potential of plant extract antioxidants corresponds to the degree of the discoloration [30]. The highest (*p* < 0.05) antioxidant activity based on DPPH (25.6 ± 0.1 µmol Trolox/g) and FRAP (46.8 ± 0.1 µmol Trolox/g) were obtained in the extract in which the highest content of total phenolics and flavonoid content were also obtained at 50% amplitude and 10 min extraction time. As shown in Table 1, by increasing the ultrasound amplitude, the antioxidant capacity was increased in all sonication times. It is well known that amplitude plays an important role in the intensification of the extraction due to its impact in cavitation. Some authors have found that high percentage of ultrasound amplitude can result in the breakage of bonds in the polyphenolic bonds [15]. However, increasing the time of sonication from 10 to 20 min resulted in a decrease of the antioxidant capacity. These results were consistent with an earlier report by Wang et al. [31], who found no increase in the total content of phenolic, flavonoid, and antioxidant activity with extraction time beyond 15 min when extracting blueberry leaves using ultrasonic extraction. It was evident that, for some plant materials, excessive extraction duration in water may cause degradation of some target compounds, resulting in reduced contents. According to the results from Muflihah et al. [32], a longer extraction time exhibited a negative effect of lower antioxidant from *Zingiberaceae* herbs, which was presumably due to the resultant prolonged heat exposure leading to the decreasing amount of targeted antioxidant compounds. The antioxidant activity of GSP extract was related to their chemical composition, primarily attributed to their richness in total phenolic content and total flavonoid content. The variation trend of FRAP values was consistent with the total phenolic contents. These results were in accordance with Hassan et al. [33], who observed that the phenolic contents of brown seaweed extract using UAE with a working frequency fixed at 42 kHz and a power of 100 W had a close correlation with FRAP antioxidant.

The relation between UAE conditions and response variables could be fit with quadratic and linear models, as follows:Total phenolics = 28.63 + 7.31T + 0.13A − 0.11T2 + 0.026A2 − 0.11TA (*p* < 0.0001, R^2^ = 0.90) 
Total flavonoids = −11.69 + 2.11T + 0.16A − 0.064T2 − 0.0056TA (*p* < 0.0001, R^2^ = 0.73)
DPPH = 36.28 − 0.87T − 0.36A + 0.025T2 + 0.0055A2 (*p* < 0.0001, R^2^ = 0.86)
FRAP = 1.83 + 3.85T + 0.73A − 0.11T2 − 0.0025A2 − 0.027TA (*p* < 0.0001, R^2^ = 0.99)
where T is extraction temperature and A is ultrasonic amplitude. All models were significant (*p* < 0.0001) with a minimum R^2^ of 0.73. The response surfaces of these variables are shown in Figure 1. The total phenolic content, total flavonoid contents, DPPH, and FRAP increased slowly with the increase of amplitude at a fixed extraction time and nearly reached a peak at the highest amplitude tested. As presented in the three-dimensional plots for antioxidant contents of DPPH and FRAP (Figure 1C,D), the extraction process variables effected the extraction of antioxidants in a similar way to the case of total content of phenolic and flavonoid. This was due to the fact that the antioxidant activities of GSP extract were closely associated with the bioactive compounds. Optimization criteria were set at maximum for all response variables. The highest values occurred with 50% amplitude and extraction time of 10.5 min, which yielded a total phenolics content of 106.5 mg GAE/g, total flavonoids of 8.54 mg CAE/g, DPPH scavenging activity of 25.6 µmol Trolox/g, and FRAP of 46.7 µmol Trolox/g. The models were verified by extraction using the optimal conditions. The actual and predicted response values were not significantly different, indicating that the models were suitable for predicting the extraction parameters within the studied range (Table 2). A total energy consumption of 24.15 kWh was also calculated based on the voltage and electrical current used by the system during processing time. Additionally, specific energy consumption was calculated based on the energy needed to obtain the unit weight of bioactive compounds [34,35]. For the optimal condition of 50% amplitude and 10.5 min extraction time, the specific energy consumption was 0.23 ± 0.01 kWh/mg GAE/g sample for the phenolic content and 2.85 ± 0.02 kWh/mg CAE/g sample for the flavonoid content.

### 3.3. Evaluation of Antioxidant and Sensory Properties of Green Soybean Milk Fortified with GSP Extracts

Bioactive compounds comprise an excellent pool of molecules for the production of nutraceuticals, functional foods, and food additives [36]. The pods of green soybean waste were collected from shelling process before the seed was ground for milk production. The GSP extracts produced from the optimized UAE were used as natural antioxidants in term of food additive for improving the oxidative stability in green soybean milk. Phenolic and flavonoid contents as well as the antioxidant activity of GSP-fortified green soybean milk are shown in Table 3. Among all samples, the 3% fortified milk sample had the highest (*p* < 0.05) total phenolic (136 ± 0.5 mg GAE/g) and total flavonoid (109 ± 0.5 mg GAE/g) content and the highest (*p* < 0.05) DPPH (176 ± 1.9 µmol Trolox/g) and FRAP (248 ± 0.3 µmol Trolox/g) antioxidant activity. In a similar study, Dabija et al. [37] revealed that the fortification of yogurt with hawthorn (*Crataegus monogyna*) extracted in increasing concentration levels (0.25, 0.50, 0.75, and 1% (*w*/*w*)) could promote the higher total phenolic content (3.46, 3.88, 4.22, 4.34 mg GAE/mL, respectively) and DPPH activity (19.23, 21.60, 32.02, and 33.38%, respectively). Lee et al. [38] also found that the increase in concentration level of *Inula britannica* flower extract for cheese fortification from 0.25 to 1.0% (*w*/*v*) caused an increase of total phenolic content and DPPH of from 54.8 to 70.8 mg GAE/g and from 53.3 to 79.1%, respectively. It was thus evident that the bioactive compounds and antioxidant activities of green soybean milk could be enhanced by pod-extracted fortification compared to milk alone.

The sensory evaluation of fortified green soybean milk was conducted by 100 untrained panelists on a 9-point structured scale, with 9 being the best and 1 the worst quality. All sensory attributes were in the range of 5–8, indicating that all formulae were at least moderately acceptable (Table 4). The addition of 3% (*v*/*v*) GSP extracts resulted in higher aroma (6.24 ± 1.6), sweetness (5.88 ± 1.6), and saltiness (5.94 ± 1.7) ratings, and a lower color (7.28 ± 1.2) rating, compared to the control formula. The lower appearance rating may be attributed to the intense green color of the product due to the addition of the GSP extract, which increased the green color, but reduced the luminosity of the milk. More intense green color was not generally well-accepted by consumer. Tamer et al. [39] reported that lemonade with 5% (*v*/*v*) green tea was rated lower in terms of color compared to control samples (0%). In addition, Farhan et al. [40] found that yogurt fortified with mint leave extracts had a lower color score than the control. Although color was directly related to consumer acceptability of the product [41], overall acceptability scores did not significantly (*p* > 0.05) differ between the 3% formula and the control. In fact, the color of the 3% (*v*/*v*) GSP-fortified milk, which was supposedly the greenest, was accepted equally to the color of the control. The panelists preferred the 3% formula the most, even more than the 2% formula. Based on the favorable sensory and antioxidant results, the 3% (*v*/*v*) GSP-fortified milk was selected for quantification of phytochemicals by HPLC.

### 3.4. Quantitative Analysis of Phytochemicals Composition

Quantification of phytochemical contents (procyanidins, quercetin, glycitein, daidzein, genistin, and linalool) in crude GSP extracts and green soybean milk with and without the addition of crude GSP extracts were determined using HPLC (Figure 2A–D). The most abundant phytochemicals in crude GSP extracts were procyanidins (0.72 ± 0.01 mg/100 g), followed by linalool (0.69 ± 0.11 mg/100 g) and quercetin (0.47 ± 0.02 mg/100 g). The procyanidin content in the crude GSP extracts in the present study was higher than lentils (0.5 mg/100 g) [42]. Compared to GSP, greater amounts of phytochemicals were observed in green soybean milk with crude extracts, especially procyanidins (3.89 ± 0.04 mg/100 g), linalool (2.79 ± 0.01 mg/100 g), glycitein (1.36 ± 0.01 mg/100 g), and quercetin (1.14 ± 0.01 mg/100 g) (Figure 3).

These results were not surprising because bean seeds are nutrient- and antioxidant-rich [43]. Hence, the green soybean milk had more content of these phytochemical groups than the pod-extracted sample. Nonetheless, the GSP extracts contained greater amounts of daidzein and genistein than the green soybean milk. Avanza et al. [8] compared the content of polyphenols from cowpea seeds and pods in the extracts of water by pressurized liquid extraction. Although the result showed a higher polyphenol content in pods than in seeds, there were remarkable differences between the analyzed flavonoid groups. Cowpea seed extracts exhibited higher content on quercetin, procyanidin, and other tetrahydroxylated flavonoids compared to pod extracts. Regarding pod extracts, gallic and ferulic acids, and o-hydroxybenzoic acid, were in greater abundant. These results might be due to the different groups of polyphenol compounds (flavonoid and phenolic acid) existing naturally in legume pods and seeds.

Procyanidins are a subclass of flavonoids found in commonly consumed foods, such as fruits, vegetables, legumes, grains, and nuts, which have attracted increasing attention due to their potential health benefits [44]. In addition to antioxidant properties, procyanidins have been reported to exhibit anticancer [45], anti-infectious, anti-inflammatory, cardioprotective, antimicrobial, antiviral, antimutagenic, wounding healing, antihyperglycemic, and anti-allergic activities [46]. Moreover, polyphenol compounds such as quercetin were reported to have neuroprotective properties attributed to their inhibiting activity against enzyme acetylcholinesterase [47]. Other polyphenols, such as genistein, daidzein, and glycitein were main phytoestrogens in the form of isoflavones. Phytoestrogens can also suppress the clinical symptoms of menopause caused by a decrease in the production of endogenous estrogen. Several studies have proven the protective effects of phytoestrogens on cardiovascular disease, which can decrease total cholesterol and improve heart function [48]. Linalool was present at rather high concentration levels in the pod extract. This kind of phytochemical is acyclic monoterpene, which is an important odorous constituent in a series of plant aromas. Linalool and linalool-rich essential oils are also known to exhibit various biological activities, such as antimicrobial, anti-inflammatory, anticancer, and antioxidant properties. In fact, several in vivo studies have confirmed various effects of linalool on the central nervous system [49]. The fortification of green soybean milk with GSP extracts enhanced both the antioxidant activity and the phytochemical content and variety of the products; thus, its increasing nutritional values.

## 4. Conclusions

The optimal conditions for the UAE of GSP extract are 50% amplitude for 10.5 min. Green soybean milk samples containing 3% GSP extract have the highest phenolic and flavonoid content, antioxidant activity, and overall acceptability compared to the control formula. Procyanidins were found to have the highest concentration level of green soy bean pod extract among five phytochemical analyses. Study results suggest that GSP extracts are a potential source of natural antioxidant, pharmaceutical, and functional ingredients in food industries.

## Figures and Tables

**Figure 1 foods-11-00588-f001:**
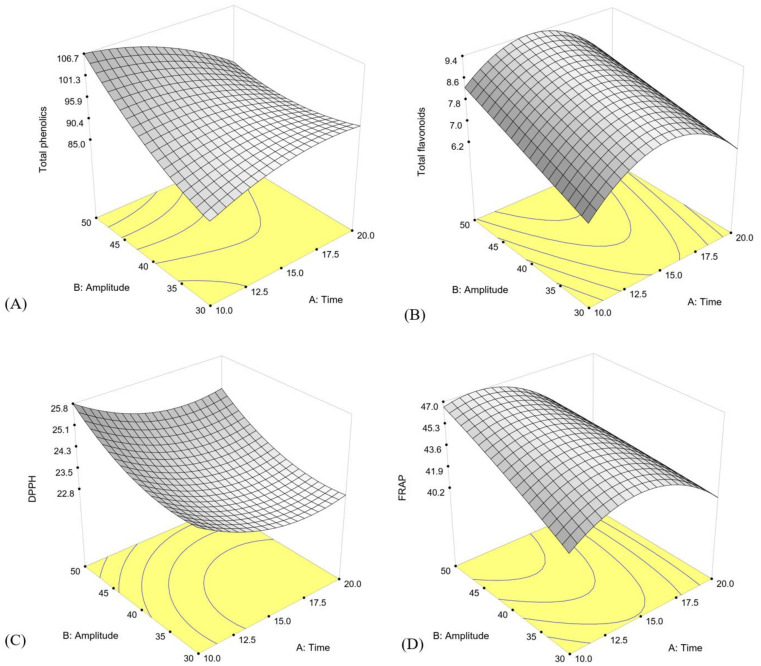
Response surfaces of (**A**) total phenolic compounds, (**B**) total flavonoids, (**C**) DPPH, and (**D**) FRAP as a function of UAE time and amplitude.

**Figure 2 foods-11-00588-f002:**
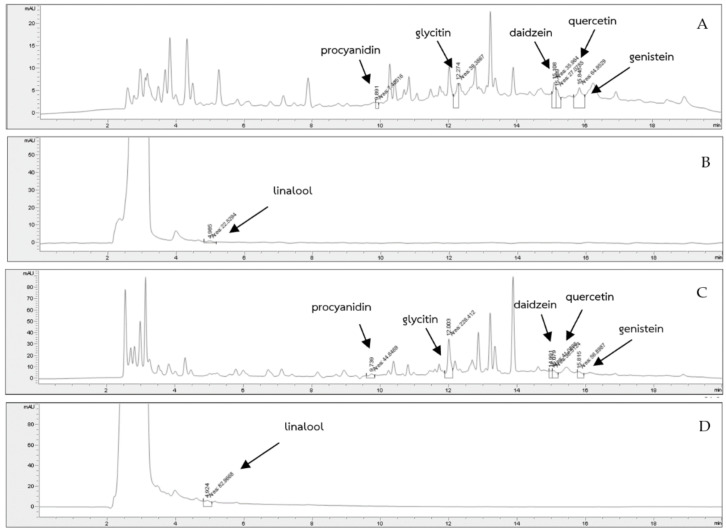
HPLC chromatogram of phenolic compounds in crude GSP extracts (**A**,**B**) and GSP-fortified green soybean milk (**C**,**D**).

**Figure 3 foods-11-00588-f003:**
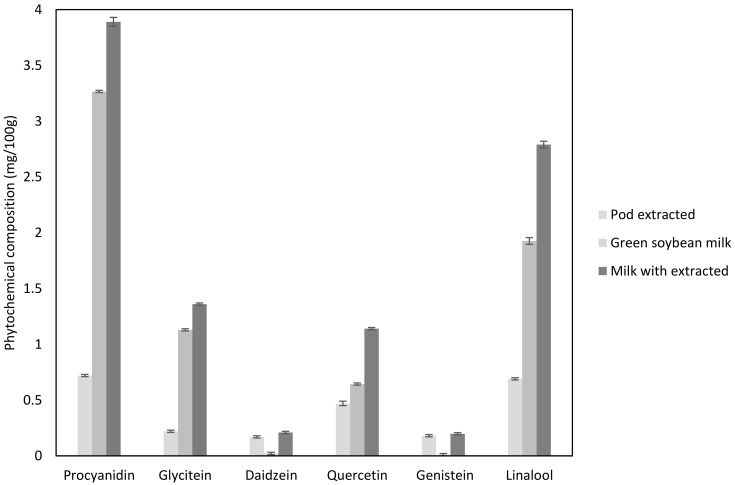
Phytochemical composition of crude GSP extracts and green soybean milk.

**Table 1 foods-11-00588-t001:** Total phenolic (mg GAE/g) and flavonoid (mg CAE/g) content and antioxidant activity based on DPPH and FRAP (%) of ultrasound-assisted green soybean pod extracts as a function of time and the ultrasonic amplitude level.

Time (min)	Amplitude (%)	Total Phenolic Content (mg GAE/g)	Total Flavonoid Content (mg CAE/g)	Antioxidant Activities (µmol Trolox/g)	
				DPPH	FRAP
10	30	85.9 ± 0.9 ^d^	6.19 ± 0.1 ^c^	24.4 ± 0.1 ^cd^	41.4 ± 0.1 ^e^
	40	91.5 ± 1.1 ^c^	6.56 ± 0.3 ^bc^	24.5 ± 0.1 ^c^	44.0 ± 0.1 ^c^
	50	107 ± 0.5 ^a^	8.94 ± 0.1 ^a^	25.6 ± 0.1 ^a^	46.8 ± 0.1 ^a^
15	30	90.9 ± 1.1 ^c^	8.50 ± 0.2 ^a^	22.8 ± 0.1 ^f^	43.4 ± 0.1 ^d^
	40	93.2 ± 1.0 ^c^	8.69 ± 0.3 ^a^	22.8 ± 0.1 ^f^	45.5 ± 0.1 ^b^
	50	103 ± 0.5 ^b^	8.75 ± 0.1 ^a^	25.2 ± 0.1 ^b^	45.5 ± 0.1 ^b^
20	30	90.6 ± 0.9 ^c^	5.75 ± 0.1 ^c^	23.0 ± 0.1 ^f^	40.4 ± 0.1 ^f^
	40	91.3 ± 1.1 ^c^	7.44 ± 0.2 ^b^	23.7 ± 0.1 ^e^	40.2 ± 0.1 ^g^
	50	90.1 ± 1.1 ^cd^	7.38 ± 0.1 ^b^	24.1 ± 0.1 ^d^	40.4 ± 0.1 ^f^

Data are expressed as means ± standard deviation (*n* = 3). Different letters (a–g) in the same column represent statistically significant difference (*p* < 0.05). DPPH = 2,2-diphenyl-1-picryl-hydrazyl radical; FRAP = ferric reducing antioxidant power.

**Table 2 foods-11-00588-t002:** Actual and predicted response values at the optimal conditions.

Responses	Predicted Value	Actual Value
Total phenolic content (mg GAE/g)	106.5	104.8 ± 2.44
Total flavonoid content (mg CAE/g)	8.54	8.48 ± 0.12
DPPH (µmol Trolox/g)	25.6	23.79 ± 0.61
FRAP (µmol Trolox/g)	46.7	45.82 ± 0.62

**Table 3 foods-11-00588-t003:** The content of phenolic and flavonoid, antioxidant activities of green soybean milk fortified with pod extract.

Pod Extract in Green Soybean Milk (%)	Total Phenolic Content (mg GAE/g)	Total Flavonoid Content (mg CAE/g)	Antioxidant Activities (µmol Trolox/g)	
			DPPH	FRAP
0 (Control)	81.3 ± 0.8 ^c^	42.0 ± 0.9 ^c^	53.2 ± 1.2 ^d^	239 ± 0.4 ^b^
1	115 ± 2.5 ^b^	85.3 ± 1.2 ^b^	125 ± 1.1 ^c^	240 ± 0.2 ^b^
2	114 ± 2.8 ^b^	85.7 ± 1.1 ^b^	132 ± 1.9 ^b^	240 ± 0.1 ^b^
3	136 ± 0.5 ^a^	109 ± 0.5 ^a^	176 ± 1.9 ^a^	248 ± 0.3 ^a^

Data are expressed as means ± standard deviation (*n* = 3). Different letters (a–d) in the same column represent statistically significant difference (*p* < 0.05).

**Table 4 foods-11-00588-t004:** Sensory analysis of green soybean milk fortified with GSP extracts. Product preference was evaluated using a 9-point hedonic scale.

Green Soybean Pod Fortification (%)	Color	Texture	Aroma	Sweetness	Saltiness	Overall
0 (Control)	7.39 ± 1.2 ^a^	6.75 ± 1.5 ^a^	6.01 ± 2.0 ^a^	5.74 ± 1.8 ^b^	5.34 ± 2.0 ^b^	6.46 ± 1.7 ^ab^
1	7.24 ± 1.1 ^ab^	6.67 ± 1.5 ^a^	6.20 ± 1.6 ^a^	6.07 ± 1.7 ^ab^	5.90 ± 1.7 ^a^	6.46 ± 1.5 ^ab^
2	7.16 ± 1.4 ^b^	6.37 ± 1.5 ^b^	6.01 ± 1.7 ^a^	6.12 ± 1.7 ^a^	5.76 ± 1.8 ^a^	6.17 ± 1.5 ^b^
3	7.28 ± 1.2 ^ab^	6.96 ± 1.7 ^a^	6.24 ± 1.6 ^a^	5.88 ± 1.6 ^ab^	5.94 ± 1.7 ^a^	6.54 ± 1.4 ^a^

Data are expressed as means ± standard deviation (*n* = 100). Different letters (a,b) in the same column represent statistically significant difference (*p* < 0.05).

## Data Availability

Data sharing not applicable.

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
