# Peer review of "Ultrasonic Extraction of Bioactive Compounds from Green Soybean Pods and Application in Green Soybean Milk Antioxidants Fortification"

_foods, 2022, doi:10.3390/foods11040588_

Round 1

Reviewer 1 Report

The authors reported extraction of bioactive compounds from GSPs and application in green soybean milk. The topic is of interest and shows several interesting findings. However, manuscript preparation and results needs to be improved.

  1. As to the UAE, how do the authors screen the single factors such as the solid/liquid ratio and ultrasonic power? Why choose the 5 g GSP powder in 100 mL water and 500 W? All these factors should be decided by a single factor experiment like extraction time and amplitudes.
  2. For data show, the author labeled every data in each table with different letters to show the statistically significant difference, they should clearly describe which groups are statistically different so that can be easily understood.
  3. Please provide the HPLC chromatogram for phenolic compounds quantification. What about the purity of each component in the extracts?
  4. Delete the "NS=non-significant" in the table notes if there is no "NS" labeled in the table.
  5. Usually P < 0.05 was considered statistically significant, why you use P <= 0.05 in your data statistical analysis?
  6. Please check the text carefully to avoid some writings that need to be rewrite, such as section 2.6.2 “Fe2+” “FeCl3.6H2O”. “Minutes” in line 218 should be “min” so as to have the same format with other units.

Reviewer 2 Report

Ultrasonic extraction of bioactive compounds from green soy bean pods and application in green soybean milk antioxidants fortification is very interesting and well written.

Presented manuscript is on good scientific level and represent a very high scientific value manuscript. 

The summary. Authors give a short presentation of manuscript.

Introduction section.

The Introduction section includes all necessary information about examined objects and problems. Formatted aim and main hypotheses are good presented at the end of Introductions' section. The problem described in manuscript is a very new and represent a new  food production and production of soy products which are a very valuable from healthy point of view.

Materials and method section is well written without any doubts.

Results. All data in Tables and Figures are good presented.

The discussion section presents a good comparison of the obtained results with other results available in the data basis.

Presented conclusions are corresponding with all information presented via Authors’ in manuscript text.

General opinion:  After carefully manuscript reading, I think, that presented manuscript is a very valuable. 

Reviewer 3 Report

Utilization of agricultural and industry generated waste and by-products has high importance according to the main concepts of circular economy. Green soybean pod is a good source of bioactive components. Development of extraction methods and increase their efficiency can contribute to achieve better yield and economy. Authors focused on the effects of process parameters of ultrasound extraction process on extraction yield, and optimization of UAE related parameters. Antioxidant activity of extract was tested in green soybean milk. Therefore, the topic of the manuscript can be considered as interesting for the readers.

The manuscript has a logical structure. Introduction section is a little superficial (see comments) but summarize the relevancies of the study. Applied analytical methods are adequate to the sample characteristics. Materials and methods are described clearly. The manuscript contains interesting and valuable results that are discussed with relevant references. Figures and tables represented well and clear the results.

Specific comments, suggestions:

Please define clearly the novelties of the study.

In the Introduction section please discuss the efficiency of UAE for extraction of bioactive components in more details (line 63-70).

Please give the average size of powdered of dried and chopped GSP.

How was the range of time and amplitude of UAE selected?

Have the authors information related to energy efficiency of UAE method (yield/input energy, for instance)?

Round 2

Reviewer 1 Report

All the comments have been addressed and the manuscript has been apparently improved.

Reviewer 3 Report

The manuscript has an interesting topic that has practical relevance, as well. Authors have revised the manuscript thoroughly according to reviewers' comments and suggestions. Amendment of Introduction and Materials and methods section, corrections in Results section, and, more detailed discussion of experimental data made the manuscript more complete and clear. The overall scientific quality of the MS has been improved significantly.